An investigation of perceptual biases in complex regional pain syndrome

http://orcid.org/0000-0002-9366-3692 De Paepe Annick L. 1 Annick.DePaepe@UGent.be
Legrain Valéry 2
Van der Biest Lien 1
Hollevoet Nadine 3
Van Tongel Alexander 3
De Wilde Lieven 3
Jacobs Herlinde 4
http://orcid.org/0000-0002-4744-8561 Crombez Geert 1
1 Department of Experimental-Clinical and Health Psychology, Faculty of Psychology and Educational Sciences, Ghent University , Ghent , Belgium
2 Institute of Neuroscience, Université Catholique de Louvain , Brussels , Belgium
3 Department of Orthopaedic Surgery and Traumatology, Ghent University Hospital , Ghent , Belgium
4 Unit of Physical Medicine, AZ Maria Middelares Hospital , Ghent , Belgium
Ferraina Stefano
Electronic publication date: 2020 Apr 2
Publication date: 2020
Volume: 8
Electronic Location ID: e8819
Received 2019 Sep 18; Accepted 2020 Feb 28
Copyright: © 2020 De Paepe et al.
Copyright year: 2020
Copyright holder: De Paepe et al.
License: This is an open access article distributed under the terms of the Creative Commons Attribution License, which permits unrestricted use, distribution, reproduction and adaptation in any medium and for any purpose provided that it is properly attributed. For attribution, the original author(s), title, publication source (PeerJ) and either DOI or URL of the article must be cited.
License URL: https://creativecommons.org/licenses/by/4.0/

Keywords: Pain, Complex regional pain syndrome, Spatial attention, Body representation

Funding: Research Foundation—Flanders, Belgium (Fonds Wetenschappelijk Onderzoek [FWO]) G.0058.11N French Speaking Community of Belgium (F.R.S.-FNRS) This study was part of a research project (G.0058.11N) granted by the Research Foundation—Flanders, Belgium (Fonds Wetenschappelijk Onderzoek [FWO]). Valéry Legrain is Research Associate at the Fund for Scientific Research of the French speaking Community of Belgium (F.R.S.-FNRS). The funders had no role in study design, data collection and analysis, decision to publish, or preparation of the manuscript.

==============================
Patients with complex regional pain syndrome (CRPS) report cognitive difficulties, affecting the ability to represent, perceive and use their affected limb. Moseley, Gallace & Spence (2009) observed that CRPS patients tend to bias the perception of tactile stimulation away from the pathological limb. Interestingly, this bias was reversed when CRPS patients were asked to cross their arms, implying that this bias is embedded in a complex representation of the body that takes into account the position of body-parts. Other studies have failed to replicate this finding (Filbrich et al., 2017) or have even found a bias in the opposite direction (Sumitani et al., 2007). Moreover, perceptual biases in CRPS patients have not often been compared to these of other chronic pain patients. Chronic pain patients are often characterized by an excessive focus of attention for bodily sensations. We might therefore expect that non-CRPS pain patients would show a bias towards instead of away from their affected limb. The aim of this study was to replicate the study of Moseley, Gallace & Spence (2009) and to extend it by comparing perceptual biases in a CRPS group with two non-CRPS pain control groups (i.e., chronic unilateral wrist and shoulder pain patients). In a temporal order judgment (TOJ) task, participants reported which of two tactile stimuli, one applied to either hand at various intervals, was perceived as occurring first. TOJs were made, either with the arms in a normal (uncrossed) position, or with the arms crossed over the body midline. We found no consistent perceptual biases in either of the patient groups and in either of the conditions (crossed/uncrossed). Individual differences were large and might, at least partly, be explained by other variables, such as pain duration and temperature differences between the pathological and non-pathological hand. Additional studies need to take these variables into account by, for example, comparing biases in CRPS (and non-CRPS) patients in an acute versus a chronic pain state.

Introduction

Complex regional pain syndrome (CRPS) is a chronic disorder associated with sensory, motor, autonomic and trophic symptoms such as pain, temperature change, skin color change, swelling and limited movement in usually one limb. CRPS often follows a minor or mild trauma to a limb (Marinus et al., 2011). The pathophysiology of CRPS is complex and still poorly understood, encompassing both structural and functional changes of the central nervous system (Juottonen et al., 2002; Krause, Förderreuther & Straube, 2006; Maihofner et al., 2003, 2004, 2005, 2007; Pleger et al., 2005, 2014). It has been shown that CRPS is associated with cognitive dysfunctions affecting the mental representation (De Vignemont, 2009), the perception and the use of the affected part of the body (Legrain et al., 2012). Patients report that moving the affected limb is slow, requires effort and involves conscious control (Galer, Butler & Jensen, 1995; Galer & Jensen, 1999). Studies have revealed that patients have difficulties with the perception of the shape and the position of the affected limb (Moseley, 2004, 2005; Lewis et al., 2007, 2010; Schwoebel et al., 2001; Turton et al., 2013). Moreover, it has been shown that CRPS is associated with a spatially-defined disruption of motor performance (Schwoebel et al., 2001; Reid et al., 2018). The limb may even feel disconnected from the body (Galer, Butler & Jensen, 1995; Galer & Jensen, 1999). Some authors have argued that CRPS has similarities to hemispatial neglect (Legrain et al., 2012; Galer & Jensen, 1999), a syndrome consecutive to a brain lesion and characterized by a deficit in perceiving and exploring stimuli at the side of space contralateral to the damaged cortical hemisphere (Corbetta, 2014; Hillis, 2016). However, the idea that the cognitive symptomatology in CRPS is akin to hemispatial neglect is contested (Legrain et al., 2012; Punt et al., 2013; Förderreuther, Sailer & Straube, 2004; Kolb et al., 2012; Reid et al., 2016; Sumitani et al., 2007; Jacobs, Brozzoli & Farnè, 2012).

Studies have shown that CRPS patients also tend to have more difficulties in perceiving somatosensory stimuli applied to the affected part of the body (McCabe et al., 2003; Maihöfner et al., 2006). For instance, Moseley, Gallace & Spence (2009) used a temporal order judgment (TOJ) task, in which participants have to judge which of two tactile stimuli applied to either hand in rapid time succession was perceived as being delivered first. They showed that CRPS patients gave priority to the processing of the tactile stimulus applied to the healthy hand at the detriment of the processing of the stimulus applied to the affected hand. Intriguingly, this pattern was reversed when the hands were crossed over the body midline, that is, the line separating the body in two equal parts according to its sagittal plane. Priority was given to the processing of the stimulus applied on the affected hand to the detriment of the processing of the stimulus applied to the healthy hand. This pattern of result seems to indicate that CRPS patients have difficulties to perceive stimuli arising at the side of space corresponding to the pathological part of the body, rather than at their affected limb itself whatever its position. Furthermore, it indicates that the observed perceptual deficits cannot be accounted for by deficits at the peripheral coding and the spinal transmission of somatosensory inputs (Schwenkreis, Maier & Tegenthoff, 2009), but involve higher order cortical mechanisms (Jänig & Baron, 2002). Conversely, Filbrich et al. (2017) used a similar TOJ task with either visual or vibrotactile stimuli and found that a bias towards the unaffected side of space was present only for the visual stimuli. Visuospatial biases in CRPS were also found in a study of Bultitude, Walker & Spence (2017) who have shown in addition that patients’ judgements about lateralized visual stimuli can be impacted by the posture of the limb, that is, if the hands are crossed or uncrossed. These latter results suggest that lateral cognitive difficulties in CRPS are not only determined by which side of the body is affected, but also depend on the actual position of the pathological limb (Legrain, 2017). Moreover, Bultitude, Walker & Spence (2017) found that the strength of the attentional bias was predicted by scores on a self-report measure of body perception distortion, but not by pain intensity, time since diagnosis or affected body site.

Apart from the lack of clarity regarding the precise nature of the cognitive deficits in CRPS patients, the specificity of these deficits for CRPS still has to be demonstrated. CRPS patients’ performance are rarely compared to those of patients suffering of other types of chronic pain conditions. For example, Frettlöh, Hüppe & Maier (2006) observed that patients with CRPS reported significantly more disownership feelings and under use of their painful limb as compared to patients with chronic pain syndromes of other origins. However nothing is described about the exact nature of these non-CRPS conditions and most importantly, whether the body is affected unilaterally or bilaterally. Similarly, despite the fact that Kolb et al. (2012) did not notice any difference between patients suffering from CRPS and non-CRPS chronic pain conditions affecting upper limbs, it is worth noting that for an important number of patients (8/20) of the non-CRPS chronic pain group both limbs were affected. Uematsu et al. (2009) compared performances of CRPS patients to those of patients with post-herpetic neuropathic pain and observed shifts of visual subjective body midline judgments only for CRPS patients. Moseley et al. (2008) found that imagined movements increased pain and swelling to the same extent in patients with CRPS as compared to non-CRPS pain patients. However, CRPS patients were on average slower to recognize their affected hand compared to their unaffected hand. This could be in accordance with studies suggesting that non-CRPS chronic pain patients have a heightened attention for bodily sensations (Vanden Bulcke et al., 2013; Peters, Vlaeyen & Van Drunen, 2000). We could therefore expect that these patients, in contrast to CRPS patients, would show a perceptual bias towards their affected limb, instead of away from their affected limb. One study of Moseley, Gallagher & Gallace (2012) contradicts this hypothesis. In this study unilateral back pain patients performed a TOJ task with tactile stimuli delivered to either side of their lower back. Similar to the results found in CRPS patients, stimuli presented on the unaffected side were prioritized in favor of those presented to the affected side. However, no CRPS patients were included in this study, so no direct comparison between CRPS and non-CRPS patients was made.

The aim of the study was to replicate the findings of Moseley, Gallace & Spence (2009) on space-based perceptual biases in patients with CRPS to clarify the exact nature of the cognitive deficits observed in this patient group and to extend these results by comparing a CRPS group with two non-CRPS pain control groups to verify whether these space-based biases are specific to CRPS patients. To this end, CRPS patients performed TOJ tasks using pairs of tactile stimuli applied to either hand and their performances were compared to those of two groups of patients with lateralized chronic pain but not from CRPS origin: a group of patients with unilateral wrist pain and a group of patients with unilateral shoulder pain. The TOJ task was performed either with the arms in a normal uncrossed posture or with the arms crossed over the body midline so that each arm laid in the opposite side of space. If lateralized cognitive deficits are not specific for CRPS patients, we expected to observe biased TOJs also in the other patient groups. In addition, given that somatosensory inputs can be spatially coded according to different reference frames, different hypotheses can be proposed. If the direction of the biases depends on an anatomical reference frame, they should affect the perception of the stimuli applied on the pathological limb whatever its position in external space. On the contrary, if the direction of the biases depends on an external space reference frame, they would not depend on limb posture and participants’ judgments should be biased towards the side of space in which their affected limb normally resides (Moseley, Gallace & Spence, 2009).

Methods

Participants

Participants aged between 18 and 70 years were recruited from January 2014 to May 2015 at Ghent University Hospital and the Ghent Maria Middelares Hospital. Three groups of participants were recruited (see criteria below): (1) patients with CRPS type I of one of the upper limbs, (2) patients with unilateral wrist pain and (3) patients with unilateral shoulder pain. Participants from all groups were included when they were native Dutch speaking and had experienced unilateral upper limb pain for longer than 3 months. They were excluded when they also reported pain at the opposite side of the body with respect to their affected limb, nerve injury (e.g., CRPS type II), ongoing limb trauma or recent (<3 weeks) surgery of the painful limb. At entry, all participants were tested with the same battery of tests to either confirm (in case of CRPS patients), or rule out (in case of wrist and shoulder pain patients) the diagnosis of CRPS according to the Budapest criteria (see Appendix). All patients had normal or corrected to normal vision. We aimed at recruiting 20 patients in every group. The study was approved by the Ethics Committee of University Hospital Ghent (registration number: 2013/706) in agreement with the Declaration of Helsinki. All participants gave their written informed consent and received a compensation.

Self-report measures

Participants completed an ad hoc questionnaire assessing socio-demographic characteristics, the Pain Grading Scale (PGS; Von Korff et al., 1992) and a Hand Dominance Questionnaire (Van Strien, 1992).

After each experimental block, a series of self-report items assessed (i) the intensity of the vibrotactile stimuli on each hand (Likert scale from 0 “not intense at all” to 10 “very intense”); (ii) to what extent they were able to concentrate during the task (Likert scale from 0 “not at all” to 10 “very well”); (iii) to what extent they experienced the task as fatiguing (Likert scale from 0 “not at all” to 10 “very much”). At the end of the experiment, additional items assessed to what extent participants (iv) directed their attention to the vibrotactile stimuli; (v) made an effort to complete the task; (vi) experienced fear/tension during the task and (vii) found the task meaningful (all measured on a Likert scale from 0 “not at all” to 10 “very much”). Participants also completed a set of self-report questionnaires (see full methods and recruitment flow charts presented here: http://hdl.handle.net/1854/LU-7179946), but these results were not used for the purpose of this study and are therefore not further discussed.

Stimuli and apparatus

Participants were seated in a dimly lit room with their hands, palms down, resting on a table (see Fig. 1). The distance between the edge of the table, near the trunk and the index fingers was 35 cm and the distance between both index fingers was 40 cm. A total of 35 cm in front of the imaginary line connecting both index fingers, a red fixation LED was positioned in the middle of this line. The participant’s head was maintained static using a chin rest. To protect them from any auditory distraction, all participants wore headphones through which continuous white noise (46 dB) resounded. The experimenter was sitting opposite to and facing the participant.

Figure 1 Experimental set-up of the TOJ task.

(A) Uncrossed arms condition. (B) Crossed arms condition.

Vibrotactile stimuli were delivered using two magnet linear actuators (C-2 TACTOR; Engineering Acoustics, Inc., Casselberry, FL, USA), attached to the sensory territory of the superficial radial nerve of each hand (10 ms duration, 200 Hz). The actuators were driven by self-developed software and a controlling device that converted electrical signals (Watt) into oscillating movements of the actuators against the skin. The intensity of the vibrotactile stimuli were determined individually and matched between both hands by means of a double random staircase procedure, based on the staircase procedure of Levitt (1971). In the first part of the procedure, 16 stimuli presented on the left hand were judged relative to a reference stimulus with maximum intensity (power = 0.21 Watt) on a 5-point Likert scale ranging from 1 (“almost no sensation”) to 5 (“maximum intensity”). The intensity that corresponded to an average rating of three was selected as the stimulus intensity for the left hand and served as the reference stimulus for the second part of the staircase procedure. In the second part, another 16 stimuli were presented, now to the right hand and were compared to the selected reference stimulus on the left hand on a 5-point Likert scale (1 = “more than less strong”, 2 = “less strong”, 3 = “equally strong”, 4 = “stronger”, 5 = “much stronger”). The intensity that resulted in an average rating of three was selected as the intensity for stimuli on the right hand.

Procedure

In the first phase of the study, participants completed the socio-demographic questionnaire, the Pain Grading Scale and the Hand Dominance Questionnaire (~10 min). In the second phase, participants were seated and the diagnostic screening (interview + testing) for CRPS took place (~20 min). In the third phase of the study (~90 min) during which the TOJ task was performed, the experimenter attached the actuators to the hands and gave the participants instructions about the staircase procedure. Following this, the headphones were turned on and the staircase procedure was initiated. Responses were inserted manually on a keyboard by the experimenter. As soon as the staircase procedure was finished, headphones were temporarily removed.

During the TOJ task, participants were instructed to fixate their gaze on a red LED in front of them, to place their chin in the chin wrest and to keep their hands still on the table throughout the task. After receiving these instructions, headphones were turned back on. The TOJ task started with three practice blocks of increasing difficulty. In the first practice block (8 trials), participants were administered only one tactile stimulus in each trial (4 left and 4 right, in random order) and were asked to locate the stimulus (“left” versus “right”) in order to practice response mapping. In the second practice block (12 trials), pairs of tactile stimuli were administered one to either hand and separated by the three largest stimulus onset asynchronies (SOA’s) used during the experiment, that is, ±200, ±90 or ±55 ms (negative values indicating that the stimulus to the left hand was applied first, positive values that the stimulus to the right hand was applied first) (Gallace & Spence, 2005). Participants had to report verbally which of the stimuli they perceived as first delivered (“left first” versus “right first”). The third practice block (18 trials) was identical to the second but was made up of 18 trials and participants were asked to cross their hands over the body midline (which arm was on top was counterbalanced). When instructions were not completely understood, or if performance was suboptimal, practice blocks were repeated until performance was satisfactory. In addition, participants could only proceed from the third practice block to the first experimental block when a minimal performance of 75% was achieved on trials with the highest SOA (±200 ms).

During the experiment proper, 4 blocks of 60 trials each were presented to the participants. Each trial was made up of a pair of tactile stimuli one administered to the left and one on the right hand, according to five possible SOAs ranging from 10 to 200 ms (De Paepe et al., 2014, 2015). The ten different trial types (±200, ±90, ±55, ±30, ±10 ms) were delivered six times each in the four blocks in random order (Gallace & Spence, 2005). The position of the arms was either uncrossed or crossed. This position was alternated between blocks and the order was counterbalanced across participants. Each trial started with the illumination of the red fixation LED for 1 s, followed by the first tactile stimuli of the pair. Participants reported verbally on which hand they perceived the stimulus as first delivered (“left hand” versus “right hand”), regardless of arm position. The experimenter inserted these responses manually on a keyboard (a = “left hand”, p = “right hand”). Participants were asked to maintain a steady pace in responding and to be as accurately as possible. After each experimental block, participants filled in the post-block items and temperature was reassessed on the back of both hands.

TOJ measures

Based on the procedure of Spence, Shore & Klein (2001), the proportion of trials on which participants reported the tactile stimulus on their painful limb first was calculated for each participant, for each SOA and for each posture (crossed versus uncrossed arm position). A sigmoid function was then fitted to these proportions and a standardized cumulative normal distribution (probits) was used to convert the proportion of left hand/right hand first responses (left hand first when the left hand/wrist/shoulder was painful, right hand first when the right hand/wrist/shoulder was painful) into a z-score. The best-fitting straight line was computed for each participant and for both postures (crossed versus uncrossed arm position) and the derived slope and intercept were used to calculate the point of subjective simultaneity (PSS) and the just noticeable difference (JND).

The PSS refers to the point at which a participant reports the two tactile stimuli (on the left and right hand) as occurring first equally often. This point can be interpreted as the SOA value that corresponds to a 0.5 proportion of left hand/right hand first responses (Spence, Shore & Klein, 2001). The PSS is calculated by taking the opposite of the intercept and dividing this by the slope, both derived from the best-fitting straight line. To simplify the interpretation, the sign of the PSS was inversed for participants with pain on the right hand/wrist/shoulder. As such, the PSS indicates how much time the stimulus on the unaffected limb had to presented before/after the stimulus on the affected limb, in order to be perceived as simultaneous. A positive PSS thus reflects biased TOJ at the advantage of stimuli applied on the affected limb and to the detriment of those applied on the unaffected hand, regardless of arm position (crossed versus uncrossed). Similarly, negative PSS reflects biased TOJ at the advantage of stimuli of the unaffected hand and to the detriment of stimuli of the affected hand.

The JND indicates the interval between both tactile stimuli (on the left and right hand) needed to achieve a 75% correct performance and as such, provides a standardized measure of the sensitivity of participants’ temporal perception. It is calculated by dividing 0.675 by the slope of the best-fitting straight line (Spence, Shore & Klein, 2001) and corresponds to the value obtained by subtracting the SOA at which the best fitting straight line crosses the 0.75 point from the SOA at which the same line crosses the 0.25 point and dividing it by 2.

Analyses

PSS values and their corresponding JND values were excluded from the analyses, if one of following criteria was not met: (1) The absolute value of the PSS values had to be smaller than twice the largest SOA (i.e., 400 ms); (2) the performance (% correct answers) of the participants for the largest SOA (i.e., 200 ms) had to be above 60% (i.e., well above chance level) in both postures (hands uncrossed versus crossed). Extremely large PSS values and low performance indicate that participants were not able to perform the task correctly even at large SOAs, where the task performance is expected to be nearly perfect. The difference in missing values between the uncrossed and the crossed position was compared using a chi-squared test for equality of proportions.

To investigate the equivalence of the average self-reported intensity for the left compared to the right hand, a repeated measures ANOVA was conducted with Hand (left versus right hand) as within-subjects’ factor and Group (CRPS, shoulder pain, wrist pain) as between-subjects’ factor1 . The same analysis was used to compare the average intensity of the tactile stimulation, as delivered by the apparatus, in Watt (W). The mean scores on the other self-report measures (see “Self-Report Measures”) were compared between the three patient groups with a one-way ANOVA.

To investigate whether there was a prioritization of stimuli on either the affected limb or the unaffected limb, one-sample t-tests were performed. For each patient group we tested if the PSS values in the crossed and uncrossed posture differed significantly from 0 ms. Next, in order to compare the PSS values across the different postures, results were analyzed using linear mixed effect models in R (lmerTest, Kuznetsova, Brockhoff & Christensen, 2017). Linear mixed effects models account for the correlations in within-subjects data by estimating subject-specific deviations (or random effects) from each population-level effect (or fixed effect) of interest (see West, Welch & Galecki (2007) for an elaboration). We chose to analyze the data with linear mixed models because it is a more subject-specific model and it allows unbalanced data, unlike the classical general linear models which requires a completely balanced array of data (West, Welch & Galecki, 2007). All models included a random intercept conditional on subject. First, a model was fitted to investigate the influence of posture (uncrossed versus crossed hands) across patient groups (CRPS, shoulder or wrist pain) on the PSS values. Posture and group as well as their interaction effect were entered to the model as fixed factors and a random subject-based intercept was added to the model. This model was not simplified as it included only the main variables of interest. Second, a model was fitted for each patient group separately to explore the potential influence of individual difference variables: pain intensity at the moment of testing (0–10), pain duration (in months), affected side (left or right) and temperature difference (see Appendix) between the affected and unaffected limb measured immediately after each block. For five participants (two CRPS patients, three wrist pain patients) the measurements of the temperature of the limbs after each block was missing. For these participants the temperature difference was imputed by the median difference of their group. Two-way interactions between the four variables (pain intensity, pain duration, affected side and temperature difference) and posture were included in the model. We attempted to simplify the model to obtain the most parsimonious model that fitted the data. To achieve this, we systematically restricted the full model based on Akaike’s Information Criterion (AIC) (Sakamoto, Ishiguro & Kitagawa, 1986). A variable was only included in the model when it decreased the AIC value with more than 2 units. Finally, a model was fitted to investigate the influence of posture (uncrossed versus crossed hands) across patient groups (CRPS, shoulder or wrist pain) on the JND values.

We did not a priori include or exclude participants based on their JND value. However, sensitivity analyses were performed for both the PSS and JND to check whether excluding participants with JND values larger than the largest SOA (i.e., ≤200) changed the results profoundly. Moreover, sensitivity analyses were performed excluding three patients who did not fulfill the research criteria of CRPS (see section “Participants” and Appendix).

The final models were fitted with REML estimation. The ANOVA table was inspected to test hypotheses about main and interaction effects. Kenward–Roger approximations to the degrees of freedom were used to adjust for small sample sizes (Kenward & Roger, 1997). The significance level was set at p < 0.05. The regression coefficients (β) and their associated confidence intervals were reported as a measure of the effect size. Raw data and R scripts are available at https://osf.io/x82wk/?view_only=7abac8c2449b4bfdb843d9a0cdce6ef0.

Results

Participants

An overview of patient characteristics is presented in Table 1 and results from the diagnostic screening can be found in the Appendix (Table A1). Although screening results were missing for four shoulder pain patients, it is very unlikely that these participants would have met the criteria for the diagnosis of CRPS as they never received the diagnosis of CRPS and also did not report pain on the upper extremities.

Table 1 Overview of patients characteristics for each patient group.

	ID	Age/sex/handedness	Location of pain	Diagnosis	Pain duration (months)	PGS
(grade score)	Other pain	Other	
CRPS	1	62/F/R	R wrist and hand	CRPS	4	3	/	Dystionia neck, familial tremor	
	2	37/F/R	R elbow, wrist, hand, pink	CRPS	18	4	R shoulder	/	
	3	66/F/R	R wrist and hand	CRPS	6	4	/	/	
	4	68/F/R	R wrist, hand, lower arm	CRPS	6	3	/	Fibromyalgia	
	5	48/F/R	L wrist and hand	CRPS	3	4	L knee, L frozen shoulder	/	
	6	45/F/R	L wrist and hand	CRPS	7	3	/	/	
	7	49/F/R	L wrist and hand	CRPS	3	1	L frozen shoulder	/	
	8	45/F/R	R hand, wrist, elbow, shoulder		18	4	R frozen shoulder	/	
	9	23/F/R	R wrist, hand, elbow	CRPS	6	3	/	/	
	10	41/F/R	R Wrist, hand, elbow	CRPS	25	3	/	/	
	11	51/M/R	R wrist and hand	CRPS	6	4	R shoulder	/	
	12	57/M/R	R wrist	CRPS	30	4	/	/	
	13	53/F/R	R wrist, hand, elbow	CRPS	3	3	Fracture R elbow	/	
	14	57/F/R	R wrist and hand	CRPS	1	4	/	/	
Shoulder	1	54/M/R	R shoulder and elbow	NS	24	4	/	/	
	2	51/M/R	R shoulder and elbow	Rotator cuff	24	2	/	/	
	3	61/M/ambi	L shoulder	Rotator cuff	60	2	/	/	
	4	53/M/L	L shoulder	NS	12	2	/	/	
	5	41/F/R	R shoulder and neck	Frozen shoulder	12	4	/	/	
	6	40/F/R	L shoulder	Frozen shoulder	48	2	/	/	
	7	64/M/L	R shoulder	NS	4	4	/	/	
	8	52/M/R	R shoulder	NS	48	4	/	/	
	9	46/F/L	R shoulder	Frozen shoulder	8	4	/	/	
	10	56/F/R	R shoulder and elbow	Frozen shoulder	24	4	Arthritis, convulsions hands	/	
	11	42/M/R	L shoulder	Frozen shoulder	25	2	Chronic low back pain	/	
	12	49/F/R	R shoulder		31	2	/	/	
	13	44/F/R	R shoulder and elbow	Frozen shoulder	24	3		Scoliosis back (no pain)	
	14	64/F/R	L shoulder and upper arm	Frozen shoulder	6	4		Hypothyroidie	
	15	59/F/R	L shoulder and neck	Frozen shoulder	7	2	/	/	
Wrist	1	58/F/R	L wrist	Malunion fracture wrist	12	1	/	/	
	2	52/F/R	R wrist	NS	6	1	/	/	
	3	40/F/L	R wrist	Wrist distortion	10	4	/		
	4	30/F/L	L wrist	NS	24	3	/	/	
	5	45/M/L	L wrist and hand	Fracture wrist	10	4	/	/	
	6	28/F/R	R wrist, elbow, hand	NS	30	4	/	/	
	7	26/F/	L wrist	Wrist distortion	10		/	/	
	8	28/M/R	L Wrist and lower arm	Fracture wrist	48	2	/	/	
	9	32/F/R	R wrist, elbow, hand	Wrist distortion	14	1	/	/	
	10	53/F/R	L wrist,
Lower arm	Tendonitis	2	2	/	/	
	11	24/F/R	L wrist	NS	9	1	Low back pain	/	
	12	59/F/R	R wrist, elbow, hand	NS	60	4	/	/	
	13	26/M/R	L wrist	NS	4	3	/	/	
	14	45/F/L	R wrist and hand	Tendonitis, ehlers danlos	111	4	L knee	/	
	15	40/M/L	R wrist	NS	10	4	/	/	
	16	51/F/R	L wrist	NS	30	4	/	/	
Note:

Age in years; F, Female; M, Male; R, right; ambi, ambidextrous; L, left; NS, Not Specified. “Hand dominance” based on Hand Dominance Questionnaire. “PGS”, Pain Grading Scale. PGS missing for 1 wrist pain patient. Remark: CRPS patient 4, 8 and 13 did not fulfill the research criteria for CRPS. Analysis were performed with and without including these patients.

Complex regional pain syndrome

CRPS patients were initially diagnosed by their medical doctor. At the beginning of the research session the Budapest criteria were assessed by the researcher (see Appendix). The presence of nerve injury (CRPS type 2) was considered as an exclusion criterion. Sixteen CRPS patients (age: M = 51.31, SD = 11.72, range = 23–68 years; 3 men, 2 ambidextrous at the moment of testing; average pain duration: M = 10.29 months, SD = 9.93) took part in this study (out of the 39 participants that were contacted, 41%). The experiment was discontinued for one participant who was unable to perform the task adequately, and another had to be excluded due to contralateral upper limb pain at the time of the experiment. Three more participants did not meet the research criteria for the diagnosis of CRPS at the time of the experiment. Analyses were first performed including these participants. Next, a sensitivity analysis was performed, excluding these participants (11 participants; age: M = 48.27, SD = 11.99, range = 23–66 years; 1 male; 2 ambidextrous; 3 left side painful; average pain duration: M = 7.45 months, SD = 7.34). See http://hdl.handle.net/1854/LU-7179946 for a more detailed overview of recruitment and inclusion.

Unilateral wrist pain

Patients with unilateral ulnar wrist pain (Nakamura, 2001; Shin et al., 2004) were invited to take part in this study. Sixteen unilateral wrist pain patients (age: M = 39.69, SD = 12.38, range = 24–59 years; 4 male; 5 left handed, 2 ambidextrous; 9 left side painful; average pain duration: M = 24.38 months, SD = 28.25) participated (out of the 46 participants that were contacted, 35%). Participants could still be excluded after the study when they reported contralateral upper body pain at the time of the experiment or when the diagnostic screening resulted in a diagnosis of CRPS. However, none of the participants had to be excluded (see http://hdl.handle.net/1854/LU-7179946).

Unilateral shoulder pain

Patients with unilateral shoulder pain, due to frozen shoulder syndrome (Lewis, 2015; Robinson et al., 2012) or rotator cuff syndrome (Beaudreuil et al., 2009; Hughes, Taylor & Green, 2008; Longo et al., 2011), were invited to participate in this study. Twenty unilateral shoulder pain patients (age: M = 52.15, SD = 7.58, range = 40–64 years; 9 male; 1 left handed, 5 ambidextrous; average pain duration: M = 21.39, SD = 16.50) took part (out of the 38 participants that were contacted, 53%). Two participants who were unable to perform the task adequately, were excluded. Three additional participants were excluded due to contralateral upper body pain at the time of the experiment. In sum, 15 participants (age: M = 51.00, SD = 8.94, range = 35–64 years; 7 men; 1 left handed, 5 ambidextrous; 6 left side painful; average pain duration: M = 23.80, SD = 17.02) were included for further analysis (see http://hdl.handle.net/1854/LU-7179946).

Self-report measures

The results of the PGS are illustrated in Table 1. The mean self-reported intensity of the vibrotactile stimuli was low (left hand: M = 2.61, SD = 2.64; right hand: M = 3.00, SD = 2.45) and did not differ significantly between both hands (F (1, 42) = 3.23, p = 0.08) across all patient groups (interaction Hand × Group: F (2, 42) = 1.88, p = 0.17). Participants reported directing their attention to a large extent to the vibrotactile stimuli (M = 7.68, SD = 2.52). They reported that they were able to concentrate well during the task (M = 7.21, SD = 1.72) and that they found the task only mildly fatiguing (M = 2.80, SD = 2.42). Participants made a large effort to complete the task (M = 7.95, SD = 1.83), reported finding the task meaningful (M = 8.02, SD = 1.42) and reported little fear/tension during the task (M = 1.91, SD = 2.11). There were no significant differences between the three patient groups.

Tactile intensities

The mean intensity (in Watt) of the tactile stimuli, derived from the staircase procedure, was not significantly different between the left and the right hand (left: M = 0.094, SD = 0.023; right: M = 0.093, SD = 0.045; F (1, 39) = 0.02, p = 0.905) for none of the three patient groups (interaction hand * group: F (2, 39) = 1.51, p = 0.233). There were also no differences in intensity of the tactile stimuli between the affected and the unaffected hand (affected: M = 0.096, SD = 0.043; unaffected: M = 0.091, SD = 0.027; F (1, 39) = 0.62, p = 0.435) for none of the three patient groups (interaction hand × group: F (2, 39) = 0.37, p = 0.691).

PSS values

Missing values

In the crossed hands posture, 10 PSS values (22%) were excluded from the analyses: two values because they did not meet criterion 1, four values did not meet criterion 2, and finally three values were excluded because they did not meet both criteria. These values belonged to three CRPS patients (21%), three shoulder pain patients (20%) and four wrist pain patients (25%). No values were excluded in the uncrossed hands posture. A chi-squared test indicated that the proportion missing values was significantly larger for the crossed than for the uncrossed posture (χ2 = 11.25, p < 0.001).

Results for all groups

PSS values for each patient group and each posture are displayed in Fig. 2. The one-sample t-tests revealed that PSS values were not significantly different from 0 for each of the three groups both in the uncrossed (CRPS: M = −0.60 (95% CI [−31.80 to 30.60]), t (13) = −0.04, p = 0.97; Shoulder: M = −10.33 (95% CI [−31.42 to 10.76]), t (14) = −1.05, p = 0.31; Wrist: M = −1.45 (95% CI [−21.06 to 18.15]), t (15) = −0.16, p = 0.88) and the crossed (CRPS: M = −32.92 (95% CI [−78.89 to 13.06]), t (10) = −1.60, p = 0.14; Shoulder: M = −7.05 (95% CI [−49.79 to 35.69]), t (11) = −0.36, p = 0.72; Wrist: M = −33.23 (95% CI [−74.57 to 8.11], t (11) = −1.77, p = 0.10) posture.

Figure 2 Individual and mean PSS values.

Individual (black) and mean (grey) PSS values per group and per posture.

The model investigating the main and interaction effect of group and posture revealed no significant effects (group: F (2, 73.76) = 0.14, p = 0.87; posture: F (1, 38.65) = 2.22, p = 0.14; group × posture: F (2, 38.83) = 0.92, p = 0.41), indicating that the PSS values did not differ significantly between the three groups and between the two postures.

Individual difference variables

CRPS patients

None of the variables (posture, pain intensity, pain duration, affected side and temperature difference) improved the fit of the model. None of the variables had a significant effect on the PSS values (all F < 1.59, all p > 0.24).

Shoulder pain patients

The final model included the main effect of posture, temperature difference and pain duration and the interaction effect between posture and temperature difference and posture and pain duration. The interaction effect between temperature difference and posture (F (1,18.44) = 5.44, p = 0.03, β = −37.69, CI [−67.40 to −7.98]) was significant. Higher temperatures for the affected versus the unaffected limb are associated with more positive PSS values in the uncrossed posture, but more negative PSS values in the crossed posture (Fig. 3). Interestingly, a paired samples t-test showed that for this patient group the affected hand had a significantly higher temperature than the unaffected hand (Δ = 0.06 (95% CI [0.008–0.12]), t (29) = 2.35, p = 0.03)2 . None of the other main or interaction effects were significant (all F < 3.18, all p > 0.09).

Figure 3 Observed PSS values in function of temperature difference and posture for the three patient groups.

The lines represent linear regression lines.

Wrist pain patients

The final model included the main effect of temperature difference and pain duration. In this model the main effect of temperature difference was significant (F (1, 18.90) = 8.51, β = 12.25, CI [4.48–20.03]), indicating that higher temperatures for the affected versus the unaffected limb are associated with more positive PSS values and thus a stronger prioritization of the unaffected limb (Fig. 3). The main effect of pain duration was not significant (F (1,11.53) = 4.39, p = 0.06, β = −0.74, CI [−1.44 to −0.05]), but there was a trend suggesting that longer symptom duration might be associated with more negative PSS values (Fig. 4).

Figure 4 Observed PSS values in function of pain duration and posture for the three patient groups.

The lines represent linear regression lines.

JND values

JND values for each patient group and each posture are displayed in Fig. 5. One CRPS patient had an extremely large JND value (−933.01) in the crossed posture. This participant was not a priori excluded from the analyses, but sensitivity analyses were performed to check whether excluding this participant changed the results (see section 3.8). The model investigating the main and interaction effect of group and posture revealed a significant main effect of posture (F (1, 35.72) = 11.48, p = 0.002, β = 156.17, CI [66.13–246.21]), indicating that participants had more difficulty with the task when their hands were crossed. The main effect of group (F (2, 69) = 1.64, p = 0.20) and the interaction effect between group and posture (F (2, 36.38) = 0.74, p = 0.48) were not significant.

Figure 5 Individual and mean JND values.

Individual (black) and mean (grey) JND values per posture and per group.

Sensitivity analyses

Exclusion based on JND values

Five patients had a JND <−200 (1 CRPS patient, 3 shoulder pain patients, 1 wrist pain patient) and were excluded from the analyses. The JND of the other patients ranged from −198.07 to −43.48.

For the PSS values, the model investigating the main and interaction effect of group and posture still revealed no significant effects (group: F (2, 63.73) = 0.28, p = 0.75; posture: F (1, 33.30) = 1.25, p = 0.27; group × posture: F (2, 33.71) = 0.76, p = 0.48). For all patient groups, the models controlling for individual difference variables were refitted. For the CRPS patients, the final model included the main effect of pain intensity, affected side, posture and temperature difference and the interaction effect between posture and temperature difference. There was a significant effect of affected side (F (1, 7.78) = 7.40, p = 0.03, β = −78.07, CI [−133.44 to −22.71]), indicating that patients with CRPS affecting the left side had significantly more positive PSS values than patients with CRPS affecting the right side of their body. The interaction effect between posture and temperature difference was marginally significant (F (1,13.98) = 4.40, p = 0.05, β = 16.52, CI [2.06–30.98]). Higher temperatures for the affected versus the unaffected hand were associated with more positive PSS values. This association was stronger in the crossed versus the uncrossed posture. None of the other effects were significant (all F < 4.60, p > 0.06). For the shoulder patients, the final model included the main effects of pain duration, posture and temperature difference and the interaction effects between temperature difference and posture and between pain duration and posture. The interaction effect between posture and temperature difference was significant (F (1, 12.74) = 5.38, p = 0.04, β = −34.64, CI [−61.39 to −7.89]), indicating that lower temperatures for the affected versus the unaffected limb are associated with more positive PSS values in the crossed posture and more negative PSS values in the uncrossed posture. Finally, the interaction effect between pain duration and posture was also significant (F (1, 7.68) = 9.62, p = 0.02, β = 4.40, CI [1.68–7.12]), indicating that longer pain duration was associated with more positive PSS values in the crossed posture, while there was no clear association in the uncrossed posture. None of the other effects were significant (all F < 3.31, p > 0.20). For the wrist pain patients, the final model included the main effects of temperature difference and pain duration. The main effect of temperature difference was significant (F (1, 21.81) = 8.62, p = 0.008, β = 14.05, CI [5.32–22.79]), with lower temperature for the affected limb versus the unaffected limb associated with more negative PSS values. The main effect of pain duration was not significant (F (1, 10.78) = 4.13, p = 0.07, β = −0.76, CI [−1.49 to −0.03]), but there was a trend suggesting that longer pain duration was associated with more negative PSS values.

For the JND values, the model investigating the main and interaction effect of group and posture still revealed a significant main effect of posture (F (1, 32.75) = 28.30, β = −65.66, CI [−89.73 to −41.60]), indicating that JND values were more negative in the crossed than in the uncrossed posture. The main effect of group (F (2, 62.78) = 0.26, p = 0.77) and the interaction effect of group and posture (F (2, 33.14) = 0.09, p = 0.91) were not significant.

Inclusion criteria CRPS

Three CRPS patients did not fulfill the research criteria of CRPS (see “Participants”) during the diagnostic screening procedure. Keep in mind that these were initially diagnosed by a clinician. Analyses were performed again excluding these participants.

The final model included the fixed effect of posture, pain duration, pain intensity and the interaction between posture and pain duration and posture and pain intensity. None of the main or interaction effects reached significance (all F < 4.41, p > 0.07).

Discussion

The goal of this study was to replicate and extend the findings of Jacobs, Brozzoli & Farnè (2012), by testing whether somatosensory impairments observed in previous studies (Reid et al., 2016; Moseley, Gallace & Spence, 2009; Moseley, Gallace & Iannetti, 2012) were specific to CRPS or whether they can also characterize other types of lateralized chronic pain. TOJ tasks were used to compare perceptual biases between tactile stimuli applied to the affected or the unaffected limbs in patients with CRPS and patients with lateralized pain in one limb from non-CRPS origins, that is, patients with either wrist or shoulder pain. Next, by asking patients to adopt different postures, that is, the limbs uncrossed or crossed over the body midline, we tested whether potential biases can be determined by either the side of space corresponding to the affected hemibody or the actual position of the limbs during the experiments. Finally, we assessed whether the difference between individuals in terms of temperature difference between the affected versus the unaffected hand and the duration of the pain, could influence TOJs in the three patient groups.

In general, the results of this study did not support a bias to tactile stimuli in patients with CRPS. First, the mean PSS values were not significantly different from zero, suggesting the absence of a consistent bias at the advantage of one of the two stimulated body parts, neither the affected nor the unaffected hand. This is in contrast with the results of previous studies (Reid et al., 2016; Moseley, Gallace & Spence, 2009; Moseley, Gallace & Iannetti, 2012). Also, for the two other groups of unilateral chronic pain patients we found no evidence for a perceptual bias. Second, there was no difference between judgements of the CRPS patients and those of the non-CRPS patients. Inspection of the individual data show that there was a substantial variability between patients, maybe hampering us to find any systematic bias towards one side of space at the group level. Whereas some patients showed more negative PSS values in the uncrossed versus the crossed posture, others showed the opposite pattern. Present data are in line with a study of Filbrich et al. (2017) who also did not find a systematic spatial bias in a tactile TOJ task with CRPS patients. Moreover, the reverse pattern of biases was also found. For instance, using the visual version of the subjective body midline judgment task, some studies found systematic deviations of the judgments towards the side of space corresponding to the affected part of the body (Uematsu et al., 2009; Sumitani et al., 2007, 2014). Those results were however not replicated by other teams (Kolb et al., 2012; Reinersmann et al., 2012). The inconsistency in the literature with respect to CRPS-related cognitive deficits might reflect substantial inter-individual differences in the cognitive symptomatology of CRPS. It has been suggested that cognitive deficits are caused by a maladaptive cortical reorganization consecutive to behavioral strategies used, even implicitly, by the patients to avoid the provocation of pain (Marinus et al., 2011). Distinct strategies across the patients might therefore differentially influence cortical changes and, as a consequence, impact patients’ behavior differently. Following this reasoning, we assessed for each patient group whether individual difference variables (pain duration, temperature differences between the affected and the unaffected hand, the side of CRPS symptoms and the intensity of the pain during testing) affected the results of this study. We found some evidence that pain duration or temperature differences between the affected and unaffected hand might play a role. It is reasonable to think that the history of the pathology might influence the potential presence of cognitive deficits. The longer duration of the pathology, the higher the probability to develop cognitive deficits affecting the perception of the pathological limb. It could therefore be argued that TOJ biases were masked in the present study by the data of the patients with more recent CRPS.

It has to be noted that there are some differences in design or patient characteristics between the present and previous studies (Reid et al., 2016; Moseley, Gallace & Spence, 2009; Moseley, Gallace & Iannetti, 2012) that may explain discrepancies in findings. First, as mentioned above, the duration of the pathology might play a role in the presence of cognitive deficits. Cognitive deficits might only be apparent after a longer duration of the pathology. Previous studies (Reid et al., 2016; Moseley, Gallace & Spence, 2009; Moseley, Gallace & Iannetti, 2012) involved more chronic CRPS patients (average duration of 30, 20 and 32 months respectively) compared to the present study (average duration of 10 months). Second, in previous studies (Moseley, Gallace & Spence, 2009; Moseley, Gallace & Iannetti, 2012) participants were cold type CRPS (the affected arm is cooler than the unaffected arm), whereas in the present study there was a mix between cold and warm type CRPS (5 patients with cold type CRPS). It has been suggested that early CRPS is associated with an increased temperature of the affected limb (warm type), while CRPS of longer duration is associated with a decreased temperature of the affected limb (cold type) (Marinus et al., 2011). The transition from hot to cold CRPS could be associated with a shift in spatially defined tactile processing (Moseley, Gallace & Spence, 2009). Third, the vibrotactile stimuli used in the present study were different from the ones used in previous studies (Reid et al., 2016; Moseley, Gallace & Spence, 2009; Moseley, Gallace & Iannetti, 2012). In previous studies, stimuli were presented by means of bone conduction vibrators, which delivered vibrotactile stimuli to the finger pads. In the present study the vibrotactile stimuli were delivered to the superficial radial nerve of each hand. Nevertheless, it is important to point out that we have used these vibrotactile stimuli in several other studies that succeeded to find biases in TOJs (Vanden Bulcke et al., 2013, 2014, 2015), using the same procedure to determine the intensity as used in the present study. Moreover, from Fig. 2 it is clear that for most participants a bias was present, but just not consistently to one side within a patient group. Fourth, verbal reports were used to report which hand was stimulated first as opposed to a foot switch. We decided to use verbal reports instead of a foot switch to make the task easier for the participants. We worked with both verbal reports as well as a foot switch in the past and were able to find biased order judgments with both response modalities (De Paepe et al., 2015). Fifth, the experimenter sat opposite to and facing the participant instead of behind the participant. Moreover, she was unblinded to the aim of the study and the patient group. However, the experimenter was blinded to the particular trial that was presented and could therefore not have influenced the results intentionally.

Conversely to the PSS values, we did observe significant results on the JND index. The JND, standing for Just Noticeable Difference, is a measure of the slope of the psychometric functions fitting participants’ performances. It reflects the sensitivity of the task and the ability of the participants to perform it (Heed & Azañón, 2014). In the present study, it reflects the minimal time interval the participants need to perform the task with 75% correct responses (see Heed & Azañón, 2014; Filbrich et al., 2017 for alternative methods to measure the slope) for alternative methods to measure the slope). The JND was significantly larger in the crossed hand posture than in the uncrossed posture, meaning that the patients needed much more time to discriminate the time order between the two tactile stimuli correctly when their arms were crossed. Such an effect of posture on the participants’ performance was present in the three groups of patients and was already shown for CRPS patients in previous studies (Moseley, Gallace & Spence, 2009; Moseley, Gallace & Iannetti, 2012) (but see Yamamoto & Kitazawa (2001) for an exception). Decreased performance during TOJ tasks with somatosensory stimuli when crossing the hands on which the stimuli are applied is a very recurrent and strong effect throughout the literature (De Paepe et al., 2015; Yamamoto & Kitazawa, 2001; Shore, Spry & Spence, 2002; Sambo et al., 2013; Azañón & Soto-Faraco, 2008; Wada, Yamamoto & Kitazawa, 2004; Crollen et al., 2017; Vanderclausen et al., in press). Such an effect is supposed to be due to a conflict between a somatotopic representation of somatosensory stimuli, that is, the ability to represent them according to which body parts are stimulated, and a spatiotopic representation, that is, the ability to represent somatosensory sensations according to where the stimulated body parts are located in external space. In other words, the crossing hand effect during somatosensory TOJ illustrates the ability of the brain to remap somatosensory inputs from an initial somatotopic or anatomical representation into a spatiotopic representation for which external space is used as reference frame (Heed & Azañón, 2014). The fact that CRPS patients do show an impaired performance during tactile TOJ when their hands are crossed, just like healthy volunteers, suggests that somatosensory remapping abilities are not affected in CRPS. Similarly, the data of three CRPS patients were disregarded in the crossed hand posture, confirming that the task was too difficult to perform for these patients in the crossed posture. In comparison, in a TOJ task with somatosensory stimuli with healthy volunteers conducted in our lab, a larger range of SOA’s was used (largest SOA 600 ms). Nevertheless, two participants also had to be excluded due to poor performance (less than 80% correct) (Vanden Bulcke & Van Damme, 2015). Future studies including a control group with healthy volunteers could directly compare the JND for healthy volunteers versus CRPS patients, to confirm that the remapping abilities for CRPS patients are similar to those of healthy volunteers.

This study has some limitations. First, a small sample of patients was tested in each group and an a-priori sample size calculation based on the study of Moseley, Gallace & Spence (2009) was not performed. Results of the “Individual difference variables” section should therefore be interpreted with caution. Future studies should calculate the ideal sample size based on the effect sizes of the present and previous studies (Reid et al., 2016; Moseley, Gallace & Spence, 2009; Moseley, Gallace & Iannetti, 2012) and should presumably use a more important sample size. Nevertheless, our sample of CRPS patients (N = 14 for the uncrossed posture and N = 11 for the crossed posture) was comparable to the sample used by Moseley, Gallace & Spence (2009), Moseley, Gallace & Iannetti (2012) (N = 10) and Reid et al. (2016) (N = 13).

Second, for several patients the task was too difficult in the crossed posture as evidenced by extremely large PSS values and consequently we were not able to use some of the data for the crossed posture. This could have artificially reduced the effect of posture. Sensitivity analyses showed that excluding these participants from the analyses did not drastically alter the results. Future studies might choose to use data-adaptive methods (Vanderclausen et al., in press; Kontsevich & Tyler, 1999) in which the tested SOAs are adapted to each participant’s own performance. This has the advantage that TOJ parameters can be measured in a valid and reliable way without probing extensively all the possible SOAs.

Finally, we did not submit an a-priori locked protocol for this study. Current guidelines within the pain field recommend to preregister the research plan before data collection (Lee et al., 2018). It should be noted that this study was conducted before the publication of these recommendations.

Conclusion

The results of this study did not support the hypotheses about the existence of systematic and specific cognitive biases affecting the ability of CRPS patients to process their affected hand. However, variability of the patients’ data was large, suggesting that other factors, such as duration of symptoms and temperature differences between the affected and unaffected limb, might play a role in the development of cognitive deficits in CRPS. This could also be the case in non-CRPS chronic pain patients, as skin temperature seemed to influence tactile TOJ in these patients. Additional studies are needed that take these variables into account by, for example, comparing biases in CRPS (and non-CRPS) patients in an acute versus a chronic pain state.

Supplemental Information

Supplemental Information 1 Diagnostic screening.

Click here for additional data file.

Additional Information and Declarations

Competing Interests

Author Contributions

Human Ethics

Data Availability

1 Note that data on the self-reported intensity of the tactile stimulation was missing for one shoulder pain patient.

2 Note that for the other two patient groups this difference was not significant (CRPS patients: Δ = 0.52, (95% CI [−0.62, 1.66], t (27) = 0.93, p = 0.36; wrist pain patients: Δ = 0.53, (95% CI [−0.31, 1.38], t (31) = 1.28, p = 0.21).

The authors declare that they have no competing interests.

Annick L. De Paepe conceived and designed the experiments, analyzed the data, prepared figures and/or tables, authored or reviewed drafts of the paper, and approved the final draft.

Valéry Legrain conceived and designed the experiments, authored or reviewed drafts of the paper, and approved the final draft.

Lien Van der Biest conceived and designed the experiments, performed the experiments, analyzed the data, prepared figures and/or tables, authored or reviewed drafts of the paper, and approved the final draft.

Nadine Hollevoet conceived and designed the experiments, authored or reviewed drafts of the paper, and approved the final draft.

Alexander Van Tongel conceived and designed the experiments, authored or reviewed drafts of the paper, and approved the final draft.

Lieven De Wilde conceived and designed the experiments, authored or reviewed drafts of the paper, and approved the final draft.

Herlinde Jacobs conceived and designed the experiments, authored or reviewed drafts of the paper, and approved the final draft.

Geert Crombez conceived and designed the experiments, authored or reviewed drafts of the paper, and approved the final draft.

The following information was supplied relating to ethical approvals (i.e., approving body and any reference numbers):

The Ethics Committee of University Hospital Ghent granted Ethical approval to carry out the study (2013/706).

The following information was supplied regarding data availability:

Data is available at the OSF: Annick De Paepe, Valéry Legrain, Lien Van der Biest, Nadine Hollevoet, Alexander Van Tongel, Lieven De Wilde, Herlinde Jacobs, and Geert Crombez. 2020. “An Investigation of Perceptual Biases in CRPS.” OSF. March 12. osf.io/x82wk.

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
