# Peer review of "An investigation of perceptual biases in complex regional pain syndrome"

_PeerJ, doi:10.7717/peerj.8819_

## Round 0.1 · original submission · Minor Revisions

I have no further comments to add to those of the reviewers. Please address their comments.

Reviewer 1 ·

Basic reporting

The article is written very clearly, is appropriately supported by literature, and includes sufficient introduction and background. The Figures and Tables are of a high quality and mostly conforms to the structure suggested in the 'instructions to authors'. The paper represents a coherent body of work.

The article fails to meet the following PeerJ standards:
- The PeerJ format requests a separate 'conclusions' section, however I think this is probably only a minor departure from the suggested structure.
- It is a requirement of PeerJ that authors provide the raw data (or a link to where it can be downloaded). I did not see where to access this - can the authors please provide it. Apologies if I missed the link.

Experimental design

The article fits within the aims and scope of PeerJ, has a well defined research question that clearly fills an identified knowledge gap, rigorously tests this question using appropriate methodology, and describes the methodology in sufficient detail.

Given that the authors aim to ‘replicate and extend’ the study of Moseley and colleagues, it is noticeable that they use a different method to determine the intensity of the tactile stimuli. The stimuli in Moseley’s study were set at 140% of the sensory detection threshold for each hand, which could potentially be of lower intensity/salience to what was used in the current study (e.g. it could be that 140% of sensory detection threshold is more equivalent to 2 on the five point scale that the current authors used). More subtle tactile stimuli could have created greater sensory uncertainty, which could be a necessary condition for spatial perception biases to manifest. It could be that when stimuli are more salient, the temporal order is more accurately perceived. The authors should consider this possibility in their discussion.

Validity of the findings

The data are mostly robust and support the conclusions drawn. However, I question whether the sample sizes of each group are sufficient for the linear mixed models analyses that the authors perform to examine whether PSS is predicted by clinical factors such as temperature difference. Any conclusions drawn from these analyses are at best speculative.

It is noticeable that no pain-free controls were tested, although this would have been an easy group to recruit. Without a group of pain-free controls, there is no way that the authors can conclude that the somatospatial remapping of people with CRPS and other pain conditions is ‘normal’. Although the different pain groups showed larger JNDs in the crossed hands condition compared to the uncrossed hand condition, all this suggests is that at least some somatospatial remapping is happening. However, it doesn’t tell us if the cost of crossing hands is comparable to what would be seen in pain-free controls performing the same task. If the authors wish to make this conclusion, then they need to add data from some pain-free controls.

Additional comments

This is overall a good manuscript and a well designed study. I suggest the following changes to improve the clarity and accuracy of the manuscript:
- In both the abstract and the introduction, the authors declare that the aim of the study was to replicate the study of Moseley et al, and to extend it by comparing perceptual biases in a CRPS group with two non-CRPS pain control groups. It would be helpful if they added to these sections an explicit statement about why / to what end they were aiming to do this.
- In the abstract the authors state that they expected biased TOJs in the three patient groups, but it is not clear from the abstract why they would expect it in the wrist and shoulder pain patients.
- The abstract seems to be missing a conclusion
- Line 27 – The term “vegetative symptoms” is not an accepted term to describe CRPS and in English can mean other things. Please change to “vasomotor” or “sudomotor” or “inflammatory” or some other descriptor per the Budapest criteria

- Line 28 – Although CRPS often follows a minor or mild trauma, the CRPS is not necessarily the result of these traumas.
- Line 39-40. The wording of this sentence is somewhat complicated – suggest ‘…akin to hemispatial neglect is contested’.
- Line 58 – Bultitude and colleagues did not replicate the study of Filbrich et al – the Bultitude study came out first. Suggest instead writing ‘…were also shown in…’. It also seems relevant to mention that the bias in the study of Bultitude et al was predicted by body representation distortion.
- Line 72 ‘an important number of’ – this is somewhat vague
- Line 75. It could be made clearer what the implications of this paragraph is.
- Lines 76-78. The authors state their aim here, but it is not clear why or to what end they wish to address this aim? The authors could make it clearer what additional insight their study would add over what has already been done.
- About lines 86 to 91. The hypotheses could be expressed more clearly. For example, it is a bit hard to follow the sentences at the end ‘according to the latter hypothesis / former hypothesis…’ It is not clear what the hypotheses would be for the CRPS groups here, and why you expect changes in the non-CRPS groups at all.
- Line 98 – did patients in the first group have CRPS affecting only one upper limb?
- Line 98 – suggest deleting the word ‘pain’ following ‘(3)’
- Line 102 – not clear what you mean by ‘had insufficiently corrected visual impairments’. How was this defined?
- Section 2.1 – why were people with CRPS Type II excluded from the study? How did the authors decide on the sample size?
- Line 129 – ‘wrest’  ‘rest’
- Line 171-172 – ‘when deemed necessary by the experimenter’. Under what conditions would the experimenter deem it necessary?
- Line 251 – temperature difference is mentioned here but doesn’t appear to occur earlier in the manuscript. The authors should explain how this is measured and when earlier in the methods section.
- Line 266 and elsewhere – the authors should provide a citation for the Budapest criteria (or refer to the appendix)
- Line 281 – it states here that CRPS was diagnosed by the medical doctor, but elsewhere the authors state that the patients were assessed for CRPS at the beginning of the research session. Please clarify.
- Line 287 – did these three patients meet the clinical diagnostic criteria for CRPS
- Section 3.1.1., 3.1.2, 3.1.3 – please add average chronicity for each of these groups, since it seems to become relevant in the discussion.
- Line 321 – reporting the task as ‘meaningful’ and ‘not having to guess’ don’t seem to equate to each other in the TOJ task. By definition, participants should feel like they need to guess for at least some of the trials (e.g. the ±10ms ones).
- Lines 360-364 and 368-374. The extent to which these significant results of the LMM analyses can be considered meaningful is questionable considering the small sample size for each group.
- Lines 421, 427, and 434. The authors classify a p=.07 as ‘marginally significant’ in one place and not significant in another. They also classify p=.09 as not significant. They should be consistent, and should adhere to the convention of classing p>.05 as simply ‘non-significant’ unless they have a good a priori justification for doing otherwise.
- Line 436 – the authors state that the goal was to replicate the study of Moseley et al, however I would argue that they are understating the importance of their study. They did not just merely replicate the study of Moseley et al, but looked to extend and add to it
- Line 448 – ‘did not succeed to evidence any data in favor of’. This is unnecessarily wordy – suggest simply ‘did not support’
-

·

Basic reporting

Excellent - it ticks all the boxes. For minor suggestions, see my full review.

Experimental design

Excellent with exception of failure to lodge an a priori protocol and undertake a power calculation on known variances. This means that the study may have been underpowered to detect effects, although the data as they are presented do not give a hint that this may be the case.

Validity of the findings

Excellent.

Additional comments

I thought this was a really well conducted and very interesting study. In attempting to replicate a finding we have replicated twice, this study did not. I think it is a really important study. I think there are some improvements that could be made to the reporting, although on the whole it is really well covered. I think there are limitations that should be mentioned. I think there are potential explanations for non-replication that could be discussed and should, therefore, trigger more discussion in the field. I actually think this study adds some more fodder to the growing impression in the field that CRPS neurology is very difficult to get clarity on.

There is at least one citation that is replicated in the reference list (ie listed twice)
citation 14 is incomplete
L.G. Moseley should be G.L Moseley – this seems to apply to all listings


Line 31 I think PeerJ readers might not know what you mean by representation. I would explain this very briefly. Also line 34 you use represent in possibly another way. I recommend taking care to be precise and consistent with these terms as they can mean various things to various people.

In this same paragraph, worth adding that studies of motor processing seem consistent with a spatially defined deficit, rather than just ‘perceving its position’ – see early work (already cited): https://www.ncbi.nlm.nih.gov/pubmed/11571225 and more recent exploration: https://www.ncbi.nlm.nih.gov/pubmed/28779873

line 71 should cite other evidence of cortical dysfunction contributing to CRPS signs and symptoms shows smaller but still present effects in non-CRPS pain: https://www.ncbi.nlm.nih.gov/pubmed/18438892 furthermore, at least one study in people with back pain replicated this spatial effect, so it is not unique to CRPS: https://www.ncbi.nlm.nih.gov/pubmed/22744662


line 98 is calling shoulder patients ‘pain patients’ intentional? If so, what is the difference between them and the wrist pain patients?

Line 122 The protocol is helpful, although it is really a more complete methods section with some results. I think the reference to it as a protocol should be changed to ‘full methods and recruitment flow charts are presented here [link]. Referring to it as a protocol implies an a priori document that is locked prior to data collection. I also think the methods section of the manuscript should include a list of all assessments undertaken, whether the order was randomised and whether any were completed prior to presentation for data collection. The results section should state how long the pre-assessments took participants to complete and what the full completion rate was on questionnaires (we have found that sensitivity of our experiments reduces with longer pre-experiment burden. This burden seems very large).

It looks like you did not do an a priori sample size calculation, which is surprising because it is seeking to replicate a previous study, so effect sizes are known.

Line 157 I am not sure what equiprobability means here. I presume you mean the stimuli were counterbalanced, not truly randomised, so that all SOAs were delivered the same number of times. Please clarify in manuscript

Line 186 typo

Line 186 was experimenter blind to stimulus? Blind to hypothesis? Blind to group? Blind to condition?

Perhaps always refer to posture as posture, not sometimes condition

Line 282 please state the results from the protocol flow chard wrt to proportion of original potential participants who were finally included.

Line 472 suggests temperature difference – PSS effect only wrist and shoulder, but results suggest also for CRPS. Please clarify.

Line 487 I think in this study, the participants are not patients.

Line 506 While I agree the sample is similar, a study to replicate a finding has good opportunity to determine required sample a priori based on effect size (although I see the variance in this study is much larger than that in previous studies). I think you should note that not performing a sample calculation is a limitation. Further, in line with current standards, that there was no a priori locked protocol for this study is an oversight. See recommendations for the pain field here: https://www.ncbi.nlm.nih.gov/pubmed/29697535 citing this paper would aid getting this standard more readily adopted in pain field.

I think the discussion of why these results contrast with previous results is underdone. I think this study is very well conducted, so the different results are important to think seriously about. Here are some issues that I think you should raise:
temperature difference between limbs. In the previous studies, participants were cold type CRPS, whereas this study seems to have a mix.
Duration – you mention this, but previous studies involved more chronic patients – how might that explain the difference?
Location of tactors – previous studies used cubes with vibrotactors inside them so the stimuli were delivered to pads of fingers. Might this contribute?
The variance is in this study is much bigger than previous, which has two interesting implications: that the sample was more heterogenous and that the methods introduced more variability. One potential contributor was the use of verbal report not foot switch, the presumably unblinded experimenter (although one might expect this to increase risk of false positive not false negative results), experimenter sitting in front of participant rather than behind, CRPS patients having some other diagnoses too.

---

## Round 0.2 · Minor Revisions

Please address these remaining minor comments:

Reviewer 1 is asking for further details to be provided.

Reviewer 2 suggests to check typos in the list of references.

Reviewer 1 ·

Basic reporting

No further comment

Experimental design

No further comment

Validity of the findings

The authors have satisfied my previous comment about the lack of sufficient sample size to make any firm conclusions about individual differences. However, there is still some work to be done with regard to my second comment about the fact that the lack of a control group limits their ability to conclude that people with CRPS have 'normal' somatospatial remapping.

It is clear that the somatospatial remapping effect is well-replicated in healthy controls in many studies, and it is also clear that the people with CRPS in this study showed a somatospatial remapping effect (larger JNDs in the crossed hand condition than uncrossed hand condition). However it is not clear if the somatospatial remapping effect that the authors found was of a similar nature and magnitude to that which would have been shown by matched control participants. Indeed, it seems that the performance of the participants with CRPS might have been more impaired by crossing hands that controls might have been - e.g. the authors state that the data of ten patients were disregarded in the crossed hands posture - is that similar to the number that would have had to been disregarded when controls are tested? Therefore, even though the CRPS patients showed a somatospatial remapping effect, you cannot conclude from these data that this was "just like healthy volunteers", nor that "somatosensory remapping abilities are not affected in CRPS". That is, it looks like CRPS patients might show greater remapping deficits than controls, but it is not possible to know because there are no control data in this study. Do the authors have data from a comparable study that they could refer to?

Additional comments

Only one additional minor comment - line 559 - suggest more “chronic CRPS patients”, since the wrist and shoulder pain patients in this study had comparable chronicity to at least one previous study

·

Basic reporting

Excellent

Experimental design

Good

Validity of the findings

High

Additional comments

I think this version is much stronger and a really important contribution to the field. I have no suggestions for improvement, but just note a few minor typo's - spelling mistakes in the reference list for example.

---

## Round 0.3 · accepted · Accept

Thank you for submitting a new version with replies to comments received.